# Diversity, Relationship, and Distribution of Virophages and Large Algal Viruses in Global Ocean Viromes

**DOI:** 10.3390/v15071582

**Published:** 2023-07-20

**Authors:** Zhenqi Wu, Ting Chu, Yijian Sheng, Yongxin Yu, Yongjie Wang

**Affiliations:** 1College of Food Science and Technology, Shanghai Ocean University, Shanghai 201304, China; wzq09988@163.com (Z.W.); chuting626@163.com (T.C.); shengyj0116@163.com (Y.S.); yxyu@shou.edu.cn (Y.Y.); 2Laboratory of Quality and Safety Risk Assessment for Aquatic Products on Storage and Preservation (Shanghai), Ministry of Agriculture and Rural Affairs, Shanghai 201304, China; 3Laboratory for Marine Biology and Biotechnology, Qingdao National Laboratory for Marine Science and Technology, Qingdao 266000, China

**Keywords:** virophage, large virus, marine, interaction, alga

## Abstract

Virophages are a group of small double-stranded DNA viruses that replicate and proliferate with the help of the viral factory of large host viruses. They are widely distributed in aquatic environments but are more abundant in freshwater ecosystems. Here, we mined the Global Ocean Viromes 2.0 (GOV 2.0) dataset for the diversity, distribution, and association of virophages and their potential host large viruses in marine environments. We identified 94 virophage sequences (>5 kbp in length), of which eight were complete genomes. The MCP phylogenetic tree showed that the GOV virophages were widely distributed on the global virophage tree but relatively clustered on three major branches. The gene-sharing network divided GOV virophages into 21 outliers, 2 overlaps, and 14 viral clusters, of which 4 consisted of only the GOV virophages. We also identified 45 large virus sequences, 8 of which were >100 kbp in length and possibly involved in cell–virus–virophage (C–V–v) trisome relationships. The potential eukaryotic hosts of these eight large viruses and the eight virophages with their complete genomes identified are likely to be algae, based on comparative genomic analysis. Both homologous gene and codon usage analyses support a possible interaction between a virophage (GOVv18) and a large algal virus (GOVLV1). These results indicate that diverse and novel virophages and large viruses are widespread in global marine environments, suggesting their important roles and the presence of complicated unknown C–V–v relationships in marine ecosystems.

## 1. Introduction

Virophages are a group of small double-stranded DNA viruses with viral particles ranging from 35 to 70 nm in diameter and genome sizes between 13 and 30 kbp [1]. They have been assigned to one family and two genera by the International Committee on Taxonomy of Viruses (ICTV) [2]. Currently, virophages have been classified into four orders and seven families [3]. A virophage cannot proliferate independently in eukaryotic host cells and must rely on the host virus’s viral factory for maturation, resulting in the morphological deformities and reduced virulence of the host virus [4,5,6].

The first laboratory co-culture isolation of Sputnik, with the protozoan amoeba *Acanthamoeba castellanii* as the eukaryotic host, led to the recognition of this particular virus as a virophage [4]. Subsequently, mavirus, isolated from the marine environment, could integrate in a specific way into the genome of the eukaryotic host of a marine phagotrophic flagellate, demonstrating the diversity of virophages [7]. The rise of metagenomics has also contributed to the discovery of various virophages. In 2011, a novel virophage, Organic Lake virophage (OLV), was assembled in the Organic Lake metagenome database [8]. Subsequently, complete genomes of new virophages were discovered in many geographically separated freshwater lakes [5,9,10,11,12], e.g., the Yellowstone Lake, and Dishui Lake. In addition to protozoan amoebae and marine flagellates, unicellular eukaryotic green algae were recently found to be eukaryotic hosts for the virophages [10,11,13].

Algae are one of the organisms that live widely on Earth, and can even grow in extreme environments. They are common in aquatic environments, including both fresh and brackish water [14]. The number of known algal species is estimated to be between 30,000 and 1 million, with the internet database AlgaeBase (http://www.algaebase.org) (accessed on 5 January 2023) describing over 150,000 species, and the number of species estimated to reach even 350 million [15]. Algae play an ecologically important role in producing the greatest amount of oxygen for living organisms, capturing carbon dioxide during photosynthesis [16], as well as organic carbon production and nutrient cycling, being the ultimate food and energy source for other organisms [17]. Increased algal biomass may cause red tides, i.e., the production of toxins leading to the death of organisms and reduced biomass, which may alter the community composition of phytoplankton and higher trophic organisms, affecting fisheries and aquaculture [18].

The widespread distribution of algae and virophages in the aquatic environments, the discovery of virophages with unknown hosts via metagenomic analysis, and the ecological importance of algae demonstrate that C–V–v (cell–virus–virophage) systems with algae as eukaryotic hosts may also be widespread and important in aquatic ecosystems.

In this study, we mined the Global Ocean Viromes 2.0 (GOV 2.0) [19] dataset for the diversity of virophages and large viruses, as well as potential C–V–v relationships. We identified abundant, novel, and mavirus-distinct virophages in the marine environments, along with diverse large algal viruses. The C–V–v relationships with algae as eukaryotic hosts emphasize the essential roles played by these viruses in the global aquatic environments.

## 2. Materials and Methods

### 2.1. Identification of Virophage Genomic Sequences

Open reading frame (ORF) prediction was performed using the Prodigal v2.6.3 [20] for sequences larger than 5 kbp in the GOV 2.0, and a local protein dataset was reconstructed for all proteins encoded by the predicted ORFs. The 371 major capsid protein (MCP) amino acid sequences of virophages [1,12] were used as query sequences. The contigs with matches to the local protein dataset (BLASTp, E-value < 10^−5^) were considered as potential virophage sequences. The proteins encoded by these virophage ORFs were extracted and compared with the NCBI nr virus database (BLASTp, E-value < 10^−5^) to identify whether there were other conserved genes such as minor capsid protein, DNA packaging ATPase, and cysteine protease on these potential virophage sequences. The polinton-like virus (PLV) sequences were removed after identification by the virophage classification software [3]. Contigs greater than 5 kb in length and containing at least two conserved genes were retained and considered as the virophage sequences and further analyzed. The completeness of the contigs was judged based on the presence of repetitive sequences at the 5′ end and 3′ end of the contigs. The presence of overlap on both ends was considered as a circular genome, and the presence of terminal inverted repeats greater than 100 bp was defined as a linear genome [5].

### 2.2. Identification of Large Virus Genomic Sequences

#### 2.2.1. Hidden Markov Models (HMMs)

The coding sequences (CDSs) of five nucleocytoplasmic virus orthologous genes (NCVOGs) [21] were downloaded from NCBI. They were DNA polymerase elongation subunit family B (NCVOG0038), D5-like helicase-primase (NCVOG0023), ATPase (NCVOG0249), DNA or RNA helicases of superfamily II (NCVOG0076), and poxvirus late transcription factor VLTF3-like (NCVOG0262). Sequence alignments were performed using the mafft v7.450 software [22] with default parameters, and the generated comparison files were used to generate model 1 using the hmmbuild command. The hmmsearch command was used to search the GOV local database with an E-value cut-off of 10^−5^ using model 1. The contigs were extracted, merged with the CDSs of the five orthologous genes of the known large viruses, aligned again, and the above steps were repeated to obtain five result files with matches to each of these five orthologous genes. Contigs containing three or more orthologous genes were considered as potential large virus sequences.

#### 2.2.2. BLASTp

The CDSs of the five downloaded NCVOGs were redundantly reduced by CD-hit v4.8.1 [23,24] (≥98% similarity). “S” was set to 0.8, and the rest of the parameters were default values. They were then used as query sequences to search against (BALSTp, E-value < 10^−10^) contigs longer than 5 kb in the GOV local database. Contigs containing three or more homologs were considered as potential genomic sequences of large viruses.

Potential large virus contigs obtained using the above two methods were combined and subjected to further verification by using ViPTree [25], Cenote-Taker 2 [26], ViralRecall [27], and geNomad [28].

### 2.3. Genomes of Virophages and Large Viruses

ORF prediction was performed using the built-in ORF finder plug-in within the Geneious v.2022.0.2 software. The parameters were set as follows: start codons of CTG, TTG, and ATG; ORF amino acid length minimum threshold of 50; and the prediction model of a standard codon (codon_start = 1). The amino acid sequences of all ORFs obtained from the prediction were compared with the NCBI nr database as the query sequences (BLASTp, E-value < 10^−5^). The potential function of each ORF was also annotated by using InterProScan (http://www.ebi.ac.uk/interpro/search/sequence-search) (accessed on 1 August 2022) and NCBI’s CD search (https://www.ncbi.nlm.nih.gov/Structure/cdd/wrpsb.cgi) (accessed on 1 August 2022) program.

### 2.4. Phylogenetic Tree

MAFFT v7.450 with the default parameters was used for the protein sequence alignment of 94 MCPs of the GOV virophages identified in this study and 371 MCPs of known virophages, and the alignments were then trimmed by using trimAl v1.2 software [29]. The FastTree v 2.1.11 [30] software was used to reconstruct phylogenetic trees (WAG model; gamma parameter estimated).

### 2.5. Gene-Sharing Network

Sixty-four GOV virophage sequences (>10 kb in length) and 2517 sequences (>10 kb in length) that were classified as *Lavidaviridae* and downloaded from the IMG/VR v4 high-confidence database were reidentified as virophages with the classification software [3]. They were then analyzed by using the vConTACT2 (v 0.11.3) [31], with none selected for the database, and the rest as default parameters. Cytoscape v3.7.2 [32] was used for data visualization.

### 2.6. Identification of Giant Virus-Specific Putative Defense Systems

All 94 GOV virophage sequences obtained in this study were broken into k-mers of 15, 20, 25, 30, 35, and 40 nt with the scripts used in our previous work [11]. They were then compared with the identified GOV large virus sequences (≥100 kb in length) using BLASTn, and the maximum number of allowed mismatches was set to 1. Repeated sequences and Cas-associated proteins were searched for upstream and downstream of the large virus genomic sequences that matched the k-mers.

The identification of Cas-related proteins was referred to in our previous work [11]. Briefly, all genes upstream and downstream of the large virus genomic sequences that matched the k-mers were predicted, and their potential functions were annotated by using InterProScan [33], eggNOG-mapper [34], Batch CD-Search [35], or HHpred [36]. Potential Cas-related proteins were further confirmed through structural comparison with known Cas proteins (SWISS-MODEL).

### 2.7. Genetic Association between the GOV Virophages and Large Viruses

A database was initially constructed, comprising all proteins encoded by the eight GOV large viruses (genome size ≥ 100 kb), and the large viruses that were attested or likely to form the C–V–v relationships [4,7,8,11,13,37,38,39,40]. The protein sequences of 94 GOV virophages were used as queries to search (BLASTp; E-value < 10^−5^, coverage ≥ 50%, and identity ≥ 30%) this local protein database for the proteins shared between the large viruses and GOV virophages. The matched proteins were then searched against the NCBI nr database (BLASTp, E-value < 10^−5^) for homologous counterparts in other large viruses and virophages.

A maximum likelihood (ML) phylogenetic tree was reconstructed as follows: The protein sequences were aligned using MUSCLE [41], and were trimmed using trimAl v1.2 [29]. The best ML tree-building model was predicted via MEGA X v10.2.6 [42].

### 2.8. Codon Usage Analysis

The codon usage frequency of large viruses and virophages, including those involved in the C–V–v systems [4,7,8,11,13,38,43], was analyzed according to the online website (https://www.bioinformatics.org/sms2/index.html) (accessed on 15 August 2022), and was visualized using a heat map.

### 2.9. Distribution of the GOV Large Viruses and Virophages

The GOV dataset [19] contains 145 sampling sites worldwide, including the Pacific Ocean, Atlantic Ocean, Indian Ocean, Arctic Ocean, Red Sea, and Mediterranean Sea, with longitudes ranging from 179.52° W to 179.141° E and latitudes ranging from 62.2231° S to 79.33349° N, and sampling depths ranging from 5 m to 4000 m. The distribution of virophages and large viruses identified in the GOV dataset was visualized according to their sampling sites.

## 3. Results and Discussion

### 3.1. 94 GOV Virophage Sequences Identified

The GOV ORFs protein database was searched (BLASTp) by using 371 known MCP protein sequences of virophages as queries. A total of 421 sequences (E-values < 10^−5^) were matched. Overall, 94 (>5 kb) of these 421 sequences were confirmed to be virophage sequences based on a sequence search against the nr virus database (BLASTp) and the virophage classification software [3]. As shown in Figure 1, 72 sequences contained at least three conserved virophage genes, 22 sequences contained two conserved virophage genes, 55 sequences contained three to four conserved genes and were >10 kb in length (accounting for 58% of all virophage sequences identified in this study), and 8 were complete virophage genomic sequences (genome sizes ranging from 20 kb to 29 kb), of which 7 were circular genomes and 1 was a linear genome (Table 1). The results indicate the presence of a rich diversity of virophages in the ocean. To facilitate subsequent studies, we numbered these 94 virophages (>5 kb) in order of sequence length, from largest to shortest.

### 3.2. Diversity of the GOV Virophages

The MCP phylogenetic tree was reconstructed to explore the diversity of these 94 GOV virophages as well as their affinities to the known virophages [1,12]. The GOV virophages occurred in all major branches of the phylogenetic tree (Figure 2), indicating a rich diversity of virophages in the marine environments. Meanwhile, the majority of GOV virophages were clustered to three large branches (Figure 2) that dominantly comprised marine virophages, which suggests that they were enriched in and adapted to the marine ecosystems. Their hosts, both large viruses and protists, are likely different from those of the freshwater virophages. Notably, the isolated marine virophage mavirus was not included in these three clades, indicating a distant relationship to the GOV virophages.

Moreover, the GOV virophages were more closely related to the alga-infecting virophages than to the protozoan (e.g., ameba, marine flagellate)-infecting virophages (Figure 2), suggesting various C–V–v relationships with algae as the potential eukaryotic hosts in marine environments.

To further understand the diversity and novelty of GOV virophages, a gene-sharing network was constructed with the vConTACT for virophage sequences >10 kb (64 GOV virophages and 2517 IMG/VR v4 virophages). As shown in Figure 3, 421 viral clusters (VCs) were observed for 2256 virophage sequences, with the GOV virophages occurring in 14 of these VCs. Of these 14 VCs, 4 were composed of the GOV virophages only. Besides this, the GOV virophages also formed 21 outliers and two overlaps, bearing low similarities to all the other virophages discovered in the global datasets. Taken together, these results indicate that marine environments harbor diverse and novel virophages that await further investigation.

Notably, compared to the >195,000 viral populations that have been identified in the GOV dataset [19], it seems that not many virophages have been discovered in marine environments (Figure 2 and Figure 3). However, given that the conserved MCPs from known virophages that are mainly found in freshwater environments were used as the bait to mine the GOV dataset, it is unlikely to trace virophages that contain highly divergent MCPs; meanwhile, it is impossible to detect virophages whose sequence fragments containing MCP genes are absent in the GOV dataset. In addition, to avoid false-positive mining results, the sequences < 5 kbp in the GOV dataset were excluded from analysis in our study. Collectively, an actual community of virophages in the ocean may be more diverse than the findings shown here.

### 3.3. Genomics of Eight GOV Virophages

The complete genomes of the eight GOV virophages all encode four conserved genes of the major capsid protein (MCP), minor capsid protein (mCP), ATPase, and cysteine protease (CP) (Figure 4). As for GOVv6, the transcription direction of MCP and mCP is reversed, which is different from the same direction observed in other virophages; and, the mCP was annotated with the HHpred solely, and was distantly related to the other virophage mCPs. Taxonomically, GOVv6 could not be assigned to the seven families in the class *Maveriviricetes* (Table 1) and thus represents a novel lineage in the virophage virosphere. Accordingly, GOVv6 likely possesses unique biological features dissimilar to the known virophages.

Functionally unknown genes, homologous to large green algal viruses and the green algae of *Pyramimonas orientalis* virus [44], *Ostreococcus lucimarinus* virus 1 [45], and *Chlorella variabilis* [46], were found in GOVv3, 4, 12, and 13 (Appendix A). This suggests that their eukaryotic hosts seem to be marine unicellular algae.

GOVv4, 7, 9, 12, and 13 encode a DNA methyltransferase gene homologous to the virophage OLV, which was hypothesized to function in reducing the endonucleolytic attack mediated by the host, large-virus OLPV [8]. Similar defense mechanisms may also occur in the C–V–v mutualistic relationship of these GOV virophages.

Three joined ORFs encoding tail proteins were detected in the GOVv3 genome (Figure 4). They belonged to the tail collar protein family of the structural components of the basal plate of the phage tail fiber [47,48], and, coincidently, a member of this family from legumes is involved in plant–microbe interactions [49]. Accordingly, these three GOVv3 ORFs might also play a role in host recognition.

Based on the virophage classification software [3], GOVv3, 7, 12, and 13 were classified as *Omnilimnoviroviridae*, and GOVv4, 9, and 26 were classified as *Burtonviroviridae*. The known aquatic virophages of YSLV1, YSLV4, YSLV6, DSLV2, OLV, QLV, and YSLV5 are affiliated with these two families, with micro-algae as the potential eukaryotic hosts [5,8,9,10,50]. Consequently, marine algae may be the eukaryotic hosts of these GOV virophages.

### 3.4. Identification of the GOV Large Viruses

Overall, 27 and 125 potential large virus sequences were found by using HMMs and BLASTp methods, respectively, and contained at least three core NCVOGs [21]. Twenty sequences were identified using both methods. These 132 non-redundant potential large virus sequences were then subjected to ViPTree analysis. Forty-five (5–50 kb: 14 sequences; 50–100 kb: 22 sequences; >100 kb: 9 sequences) were clustered with known large unicellular eukaryotic viruses and belonged to the class *Megaviricetes*, as confirmed based on Cenote-Taker 2, ViralRecall, and geNomad. These 45 sequences were divided into two categories of *Mimiviridae* -related and *Phycodnaviridae* -related large viruses (Figure 5). The GOV large viruses were clustered together with known giant viruses, but appeared to be more closely related to each other (Figure 5). The eight large virus sequences greater than 100 kb in length (GOVLV1-4, 6-9) (Figure 6) may contain more genomic information, and were used for the subsequent investigation of virophage–large virus interactions.

### 3.5. Genomics of Eight GOV Large Viruses

The results of ORF annotations are summarized in Table 2. Two hundred and thirty-one ORFs of GOVLV1 (98.2%) were best matched to those of CeV (Appendix A), with a sequence similarity of 79.7–100%; GOVLV1 and CeV were clustered together on the ViPTree with a branching scale close to 0.5 (Figure 5). Apparently, CeV and GOVLV1 are the closest relatives. CeV infects the microalga (the haptophyceae *Haptolina ericina*, formerly *Chrysochromulina ericina*) [51], which, therefore, is highly likely the host of GOVLV1 as well.

About 61% of the GOVLV3 ORFs, homologous to viral genes, matched those of the large viruses infecting green algae of the family *Bathycoccaceae* (Appendix A). The closest homolog of GOVLV3 ORF56 (E-value < 10^−23^; coverage > 95%; identity > 29%) was from *Ostreococcus tauri* of the family *Bathycoccaceae*, suggesting a horizontal gene transfer that likely occurred between GOVLV3 and its potential green alga host.

GOVLVs of 2, 9, 4, 6, 7, and 8 shared the greatest number of homologous genes (68.8–100% of the ORFs that matched viral genes), with algae-infecting large viruses of CeV, PgV, and OLPVs (Appendix A), and GOVLVs of 2, 3, 4, 7, and 8 shared homologous genes with algae (Appendix A).

On the ViPTree (Figure 5), similar to the above results from the genomic analysis, the GOV large viruses were all clustered with known large algae-infecting viruses, further supporting their preying on the marine algae.

### 3.6. Interactions of the GOV Large Viruses and Virophages

To give insights into the potential associations between the GOV large viruses and virophages, homologous genes shared between the eight GOV large viruses (>100 kb) and 94 GOV virophages (>5 kb) were analyzed via BLASTp (E-value < 10^−5^; coverage ≥ 50%; identity ≥ 30%). A 2OG-FeII Oxy super family protein was shared by GOVv34 and two GOVLVs (6 and 7) (E-value < 10^−120^; coverage ≥ 98%), and a ferritin-like super family protein was shared by GOVv18 and five GOVLVs (1–3 and 8–9) (E-value < 10^−20^; coverage ≥ 97%). On the phylogenetic trees reconstructed by using close homologs of these two proteins, GOVv34 was grouped with GOVLV6 and 7 (Figure 7A), and GOVv18 was grouped with GOVLV1, 2, 8, and 9, but was distantly related to GOVLV3 (Figure 7B). Meanwhile, GOVLV1, 2, 8, and 9 were close relatives and formed a clade with the alga-infecting *Mimiviridae*-related viruses and other GOVLVs (Figure 5). Collectively, these results suggest that an interaction might occur between these GOV virophages and large algal viruses.

Meanwhile, codon usage frequencies were analyzed to look for more evidence shedding light on the potential relationships between the GOV virophages and large viruses. Similar to Sputnik-Mamavirus (v–V), GOVvs (9, 18, and 42)-GOVLV1, GOVv45-GOVLVs (6, 7, and 9), and GOVvs (12, 13, 50, and 55)-GOVLV2 were clustered together, respectively (Figure 8), suggesting a potential interaction between them.

The results of both the shared homologous gene and codon usage frequency support the potential interaction between GOVLV1 and GOVv18. Since GOVLV1 was closely related to CeV (Figure 5), which infects the golden alga *Haptolina ericina*, a C–V–v threesome comprising *Haptolina* alga-GOVLV1-GOVv18 likely exists in the ocean.

The giant virus-specific putative defense systems, identified in mamavirus and DSLLAV1 [11,52], were not detected in the GOVLVs, which might result from the incomplete genomic sequences analyzed. Alternatively, only a small number of large/giant viruses could utilize giant virus-specific putative defense systems to protect them from the parasitizing of virophages.

### 3.7. Distribution of the Virophages and Large Viruses in the Marine Environments

Virophages were detected in the Arctic Ocean, Mediterranean Sea, Red Sea, Indian Ocean, North Atlantic Ocean, South Atlantic Ocean, and South Pacific Ocean (Figure 9), indicating a wide distribution of diverse virophages in the global ocean. By contrast, the large algal viruses were found in the Arctic Ocean, South Atlantic Ocean, North Atlantic Ocean, Red Sea, and Indian Ocean (Figure 9). The wide distribution of virophages and large viruses in the global marine environments suggests their importance in the marine ecosystems in addition to the freshwater.

Most (82.2%) of the virophages and large viruses identified in the GOV were from the Arctic Ocean. Coincidently, the co-existence of virophages and large viruses in the same sampling sites was mainly found in the Arctic Ocean as well. Although the water temperature is about −1 °C to −1.7 °C from the surface to a depth of 100–225 m throughout the year in the Arctic Sea, the algae thrive well since they possess antifreeze proteins that protect them from cold temperatures [53,54]. Accordingly, large algal viruses and their parasitic virophages appeared to be enriched in the Arctic Sea. Due to the bias of the datasets, the co-existence of the virophages and large viruses did not necessarily mean that they interacted with each other. However, undoubtedly, the C–V–v systems function in the Arctic Ocean and may be the critical players in the ecosystems of the Arctic Sea water.

## 4. Conclusions

Diverse virophages were identified in the global marine environments and were mainly affiliated with *Omnilimnoviroviridae*, *Sputniviroviridae*, and a new family in *Maveriviricetes*. Virophages belonging to the family *Maviroviridae* were not found in the GOV datasets, although *Maviroviridae* contain the first virophage of mavirus isolated from coastal seawater [3,7]. The identified large viruses are closely related to the alga-infecting *Mimiviridae*- and *Phycodnaviridae*-related viruses. The potential C–V–v relationships with algae, e.g., golden algae, as eukaryotic hosts were detected and likely dominate in the marine environments, especially in the Arctic Ocean. Our findings highlight the important contributions of the C–V–v systems to the evolution and ecology of the marine environment.

## Figures and Tables

**Figure 1 viruses-15-01582-f001:**
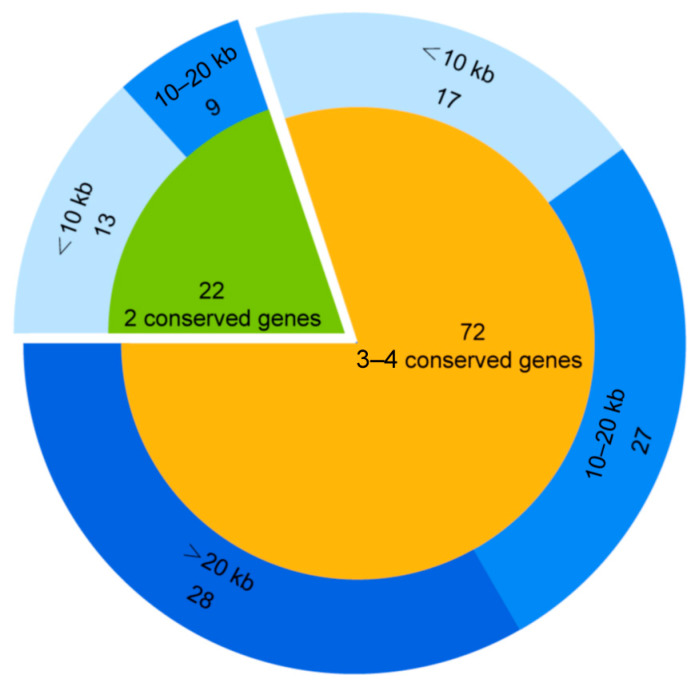
GOV virophage sequences identified.

**Figure 2 viruses-15-01582-f002:**
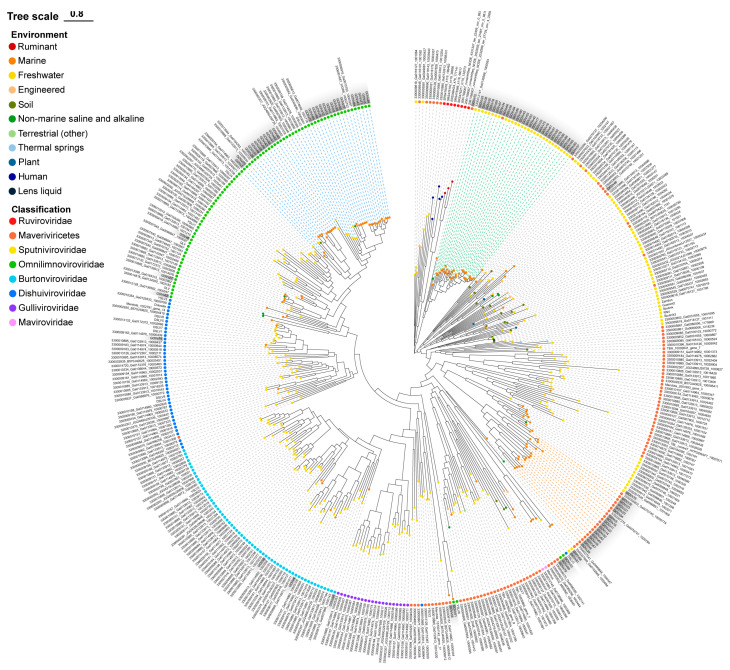
The MCP phylogenetic tree of the GOV virophages. The GOV virophage sequences are marked in grey; blue, green, and orange dashed lines indicate the three major clades of GOV virophages. The colorful dots on the top of the branches are labeled for the sampling source of the sequences, and the colorful dots near the sequence names represent the taxonomic ranks of the sequences. DSLV, Dishui Lake virophage; OLV, Organic Lake virophage; RNV, Rio Negro virophage; YSLV, Yellowstone Lake virophage; QLV, Qinghai Lake virophage.

**Figure 3 viruses-15-01582-f003:**
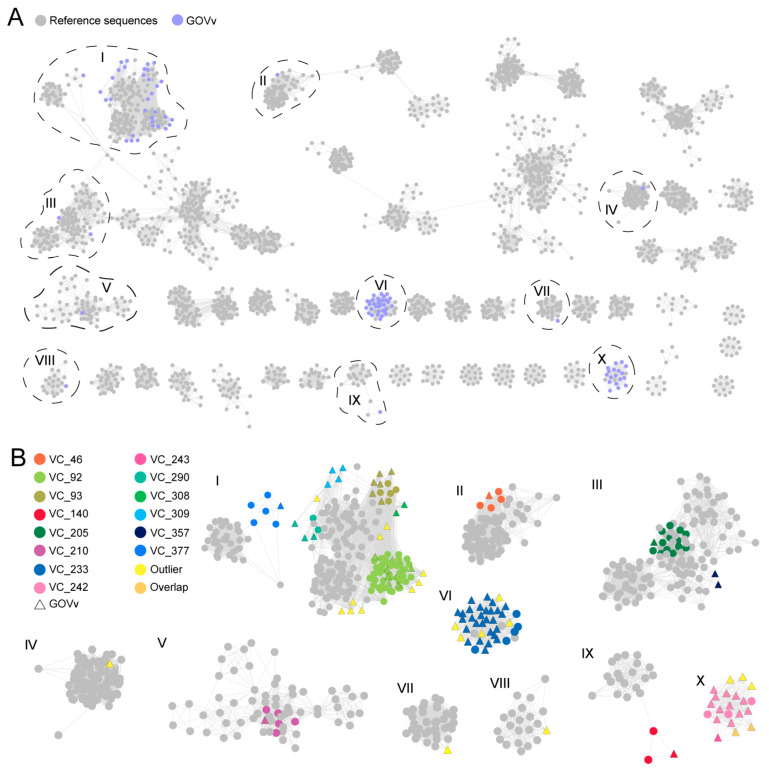
The gene-sharing network of the GOV virophages. (**A**) The GOV virophage sequences (>10 kb) are marked in purple, and the reference sequences (>10 kb) classified as *Maveriviricetes* in the IMG/VR v4 database are marked in grey. The VCs where the GOV virophages were grouped are indicated with black dashed lines and the Roman numerals. (**B**) The VCs in which the GOV virophages were clustered are shown in the magnification. The grey color represents the reference sequences in the IMG/VR database, the different colors represent the different VCs, and the triangles are marked for the GOV virophages.

**Figure 4 viruses-15-01582-f004:**
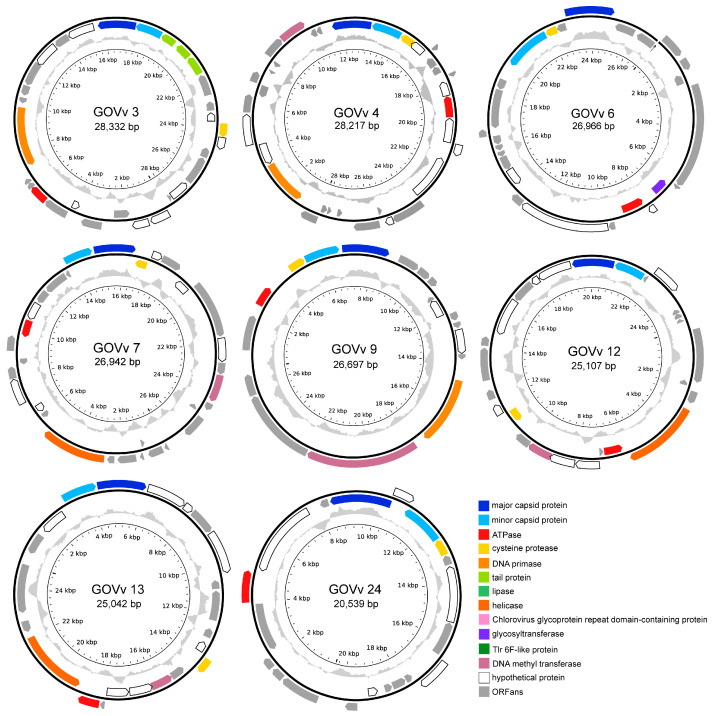
Physical map of the complete genomes of eight GOV virophages. ORFs in different colors represent different functional categories. Inner zigzag grey line denotes GC content. The linear genome of GOVv6 is shown in open circle, and the black arrow points to the opening.

**Figure 5 viruses-15-01582-f005:**
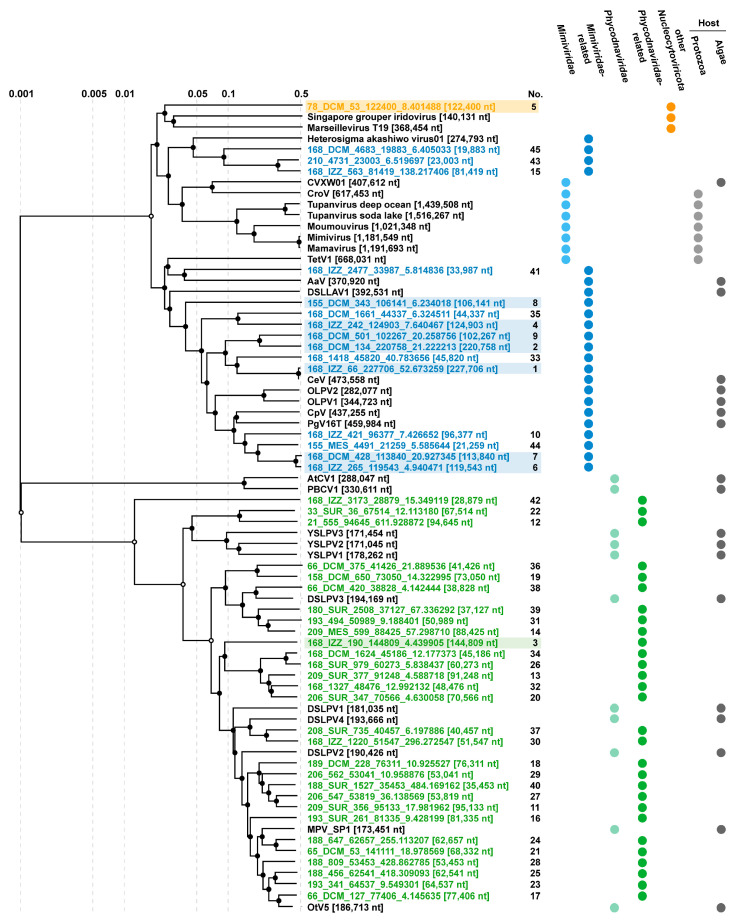
ViPTree of 45 GOV large viruses (incomplete genomic sequence > 5 kbp). The colors indicate the GOV large virus sequences. Blue indicates the *Mimiviridae* -related sequences, green indicates the *Phycodnaviridae* -related sequences, and orange indicates other large viruses. Sequences with a background color are those larger than 100 kb in length and the numbers represent sequences numbered in descending order according to their length. CVXW 01, Chlorella Virus XW01; CroV, *Cafeteria roenbergensis* virus; Moumouvirus, *Acanthamoeba polyphaga* moumouvirus; Mimivirus, *Acanthamoeba polyphaga* mimivirus; Mamavirus, *Acanthamoeba castellanii* mamavirus; AaV, *Aureococcus anophagefferens* virus; DSLLAV, Dishui Lake large alga virus; CeV, *Chrysochromulina ericina* virus; OLPV, Organic Lake phycodnavirus; CpV, *Chrysochromulina parva* virus; PgV, *Phaeocystis globosa* virus; AtCV, *Acanthocystis turfacea chlorella* virus; PBCV, *Paramecium bursaria chlorella* virus; YSLPV, Yellowstone Lake phycodnavirus; DSLPV, Dishui Lake phycodnavirus; MPV, *Micromonas pusilla* virus; OtV, *Ostreococcus tauri* virus.

**Figure 6 viruses-15-01582-f006:**
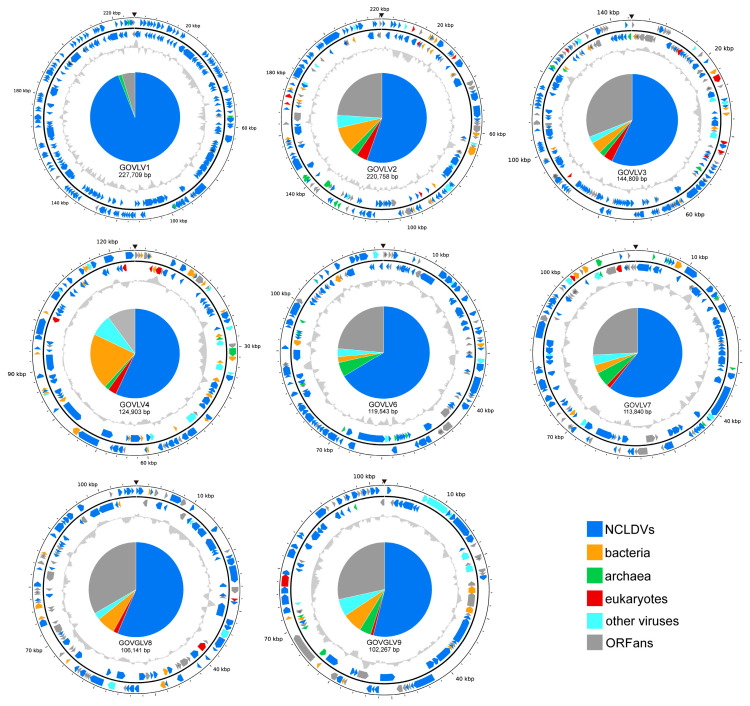
Genomic physical map (partial) of the eight GOV large viruses (sequence longer than 100 kb). ORFs in different colors represent different taxonomic categories of BLASTp top hits. Their proportions are shown in the pie chart in the center. Inner zigzag grey line denotes GC content. The black triangle indicates the break site of the genomic sequences.

**Figure 7 viruses-15-01582-f007:**
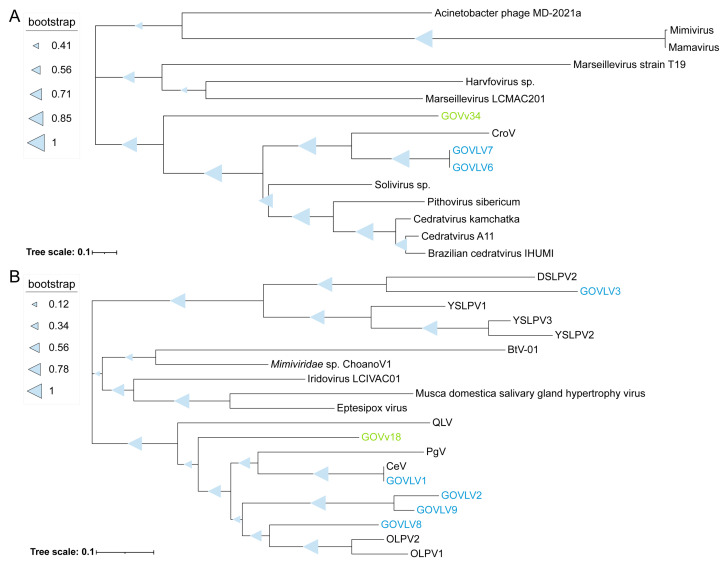
Phylogenetic tree of the genes shared between the GOV virophages and large viruses. Protein sequences were used for analysis. (**A**) GOVv34, GOVLV6-7, and other large viruses. (**B**) GOVv18, GOVLVs (1–3 and 8–9), and other virophages and large viruses. The GOV virophages are marked in green, and the GOV large viruses are marked in blue. Full names of viruses are consistent with those provided in the Figure 5 legend.

**Figure 8 viruses-15-01582-f008:**
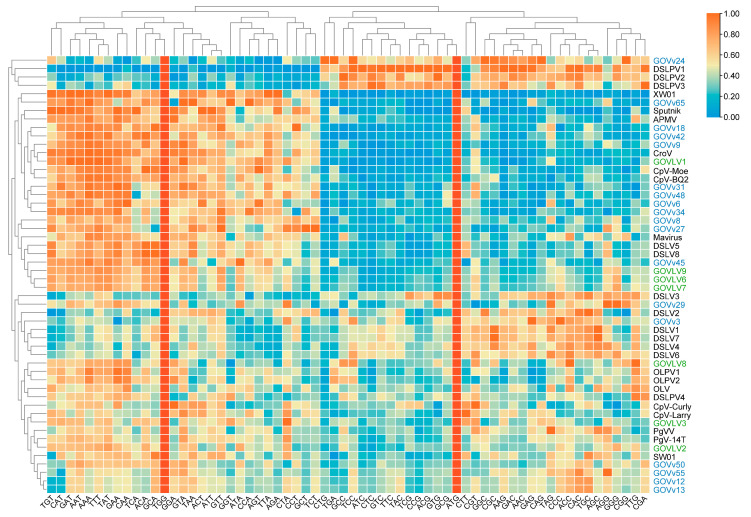
Codon usage frequencies and preferences of the GOV virophages and large viruses. Columns show codon usage frequency of each given genome, and rows represent different viruses. Full names of viruses are consistent with those provided in the Figure 2 and Figure 5 legends.

**Figure 9 viruses-15-01582-f009:**
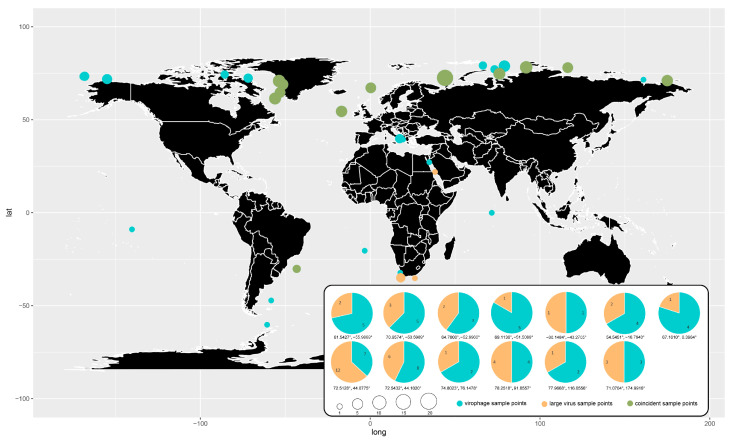
Global geographical distribution of the GOV virophages and large viruses. Blue dots represent virophages, orange dots represent large viruses, and green dots represent co-existence of virophages and large viruses. The size of the dots represents the number of total virophage and large virus sequences found at the sampling sites. The pie chart shows the percentage of the number of virophage and large virus sequences at the co-existing sampling sites.

**Table 1 viruses-15-01582-t001:** The GOV complete virophage genomic sequences.

Name	Size (bp)	Type	GC Content (%)	Number of ORF	Classification [20]
GOVv3	28,332	Circular	41.4	30	*Omnilimnoviroviridae*
GOVv4	28,217	Circular	47.9	33	*Burtonviroviridae*
GOVv6	26,966	Linear	30.7	25	*Maveriviricetes*
GOVv7	26,942	Circular	30.7	31	*Omnilimnoviroviridae*
GOVv9	26,697	Circular	28.9	19	*Burtonviroviridae*
GOVv12	25,107	Circular	39.7	26	*Omnilimnoviroviridae*
GOVv13	25,042	Circular	39.7	22	*Omnilimnoviroviridae*
GOVv24	20,539	Circular	49.8	20	*Burtonviroviridae*

**Table 2 viruses-15-01582-t002:** ORF annotations of eight GOV large viruses.

Large Virus	Genome Length (kb, Partial)	Annotated ORF (%)	The Best Matches From			
			Virus	Bacterium	Archaeon	Eukaryote
GOVLV1	227,706	95.2	233	3	-	-
GOVLV2	220,758	76.0	105	16	5	7
GOVLV3	144,809	68.1	126	8	4	7
GOVLV4	124,903	89.8	83	26	2	4
GOVLV6	119,543	76.6	98	3	7	-
GOVLV7	113,840	75.0	85	4	7	2
GOVLV8	106,141	66.7	74	8	-	2
GOVLV9	102,267	71.6	61	7	4	1

## Data Availability

The data presented in this study are openly available through iVirus.

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
