# Peer review of "Diversity, Relationship, and Distribution of Virophages and Large Algal Viruses in Global Ocean Viromes"

_viruses, 2023, doi:10.3390/v15071582_

Round 1
Reviewer 1 Report
In this manuscript, the authors have identified virophages in the global ocean virome dataset, and performed a comparative phylogenetic and gene content analysis of these elements. The research presented will be important for understanding the diversity and host-affiliation of virophages in the global ocean.
Given the global sampling approach, it is surprising to see so few virophage elements in the GOV dataset given that the GOV dataset has >195,000 viral populations! (Gregory A et al, Cell, 2019) . This is certainly not a criticism of the paper, but the authors might want to discuss why this is the case. Are we missing most of the diversity of the virophages in the ocean, or their diversity is really low in the ocean?
Specific comments:
a) The authors have used 'CRISPR-like' to refer to a putative defense system in giant viruses. I humbly disagree with this. There has been much debate regarding the fact that this system doesn't likely work in a manner analogous to CRISPR (see: doi: 10.1007/s12250-016-3801-x). I think it is better if the authors simply refer to this system as a ' giant virus-specific putative defense system'.
b) For identification of large virus sequences, the authors have used information regarding the presence of genes homologous to several core viral genes. This approach alone can produce potentially false positive results, because homologs of some of these genes are quite common in bacterial or eukaryotic sequences and bacteriophages. To reduce these false positives, the authors should further screen these candidate sequences using tools that are specific for large viruses. For example, ViralRecall (https://www.mdpi.com/1999-4915/13/2/150)
c) Figure 6: I think these are likely partial genomes/genome fragments of large viruses. They are not complete viral genomes. The authors should specify this in the figure caption.
Reviewer 2 Report
Using public available data sets, the authors identify many more (94) and very diverse virophages that exist in marine environments by bioinformatic techniques. They also discovered many (45) more large or giant viruses that are predicted to infect eukaryotic algae. However, potential genomes greater than 100 kb were only discovered in 8 of these large viruses. They suggest that at least some of these algal infecting viruses are infected by some of the virophages. The work is carefully done and represents another example of how diverse virus populations are in marine environments.
My major criticism is that I had trouble seeing the symbols and names in the figures.
Figure 2. What is the yellow line that is present on the inner circle? They define the colors blue, green and orange but not yellow. Also is it possible to enlarge the symbols in the upper right hand corner? Maybe move them to the other side of the figure. I have the same issue with Figure 3.
Figure 8. The paragraph under figure 8 refers to several viruses, e.g., GOVs 12, 13, 50 and 55 in Fig. 8 and I cannot find them.
Table 2. Replace / with -.
minor edits.
page 2, line 25. change play to played
references. Some of the references have the first letter in each word in capitals and some do not. be consistent.
The English is fine.
Round 2
Reviewer 1 Report
The authors have addressed my concerns adequately. Thank you.